# Selenium in Infants and Preschool Children Nutrition: A Literature Review

**DOI:** 10.3390/nu15214668

**Published:** 2023-11-03

**Authors:** Małgorzata Dobrzyńska, Katarzyna Kaczmarek, Juliusz Przysławski, Sławomira Drzymała-Czyż

**Affiliations:** Department of Bromatology, Poznan University of Medical Science, Rokietnicka 3 Street, 60-806 Poznan, Poland; mdobrzynska@ump.edu.pl (M.D.); katarzynakaczmarek.official@gmail.com (K.K.); jprzysla@ump.edu.pl (J.P.)

**Keywords:** selenium deficiency, selenium-rich product, infant nutrition, preschool children nutrition

## Abstract

Selenium (Se), an essential trace element, is fundamental to human health, playing an important role in the formation of thyroid hormones, DNA synthesis, the immune response, and fertility. There is a lack of comprehensive epidemiological research, particularly the serum Se concetration in healthy infants and preschool children compared to the estimated dietary Se intake. However, Se deficiencies and exceeding the UL have been observed in infants and preschool children. Despite the observed irregularities in Se intake, there is a lack of nutritional recommendations for infants and preschool children. Therefore, the main objective of this literature review was to summarize what is known to date about Se levels and the risk of deficiency related to regular consumption in infants and preschool children.

## 1. Introduction

Selenium (Se) is a naturally occurring element essential to human health in trace amounts but is harmful in excess. However, there is a narrow range between dietary deficiency and toxic levels, and thus it is necessary to carefully control Se intake, especially in infants and children [1,2].

Globally, serum Se concetration is highly variable and dependent on the environment and eating habits [3,4]. Few epidemiological studies have reported human serum Se concentrations and dietary intake estimates and the frequency of global Se deficiency or excessive consumption has not been determined. However, Se deficiency is a health concern in some areas of China [5,6], Africa, especially sub-Saharan African countries such as Ethiopia or Malawi [3,7,8], and the Andean regions of South America [9]. Se deficiency has also been observed in Hungary [4], Switzerland [10], Poland [11], and some areas in Russia [12].

Furthermore, premature infants and patients fed only with parenteral and enteral nutrition without Se supplementation are particularly vulnerable to Se deficiency [13]. Additionally, children with food allergies, phenylketonuria, or other diet-related diseases may be at risk of Se deficiency due to their diet being restricted from many Se-rich products [14,15].

Excessive selenium intake by children and infants is rare [16,17] and often caused by excessive consumption of Se supplements or Brazil nuts [17,18]. According to the European Food Safety Authority (EFSA), there is no reported risk with the current levels of Se intake in European countries from food (excluding food supplements) in toddlers and children, and selenium intake arising from the natural content of foods does not raise reasons for concern [19].

We believe that there is a risk of Se deficiency in infants and preschool children related to their daily consumption. Therefore, the main objective of this literature review was to summarize what is known to date about Se levels and the risk of deficiency related to regular consumption in infants and preschool children.

### Methodology

Relevant articles regarding Se in the diet of infants and children were retrieved from PubMed, Scopus, Web of Science, Embase, and Cochrane Library using the following keywords: “selenium” and “food products” and “healthy children” or “selenium” and “food products” and “healthy infants”. An additional search was performed for selenium content” and “healthy children” or “selenium content” and “healthy infants”. Duplicate articles and studies in languages other than English or adults were excluded.

Only studies on infant and preschool children (age: birth–5 years) were included in the analysis. The analysis took into account the type of study, country, age group, gender, demographic characteristics, Se intake, sources of Se in the diet, and serum Se levels.

All available studies were included in the analysis, without time limits, theoretically since 1973, but after narrowing the search with exclusion terms, 18 manuscripts were finally included, 5 of which were before 2000 (one work from 1982, 1995, 1996, 1998, and 1999, respectively). The reference lists of retrieved articles were screened manually to find the potentially relevant literature (Figure 1). 

## 2. Selenium in the Human Body

### 2.1. Main Function of Selenium in the Human Body

The biological functions of Se depend on its chemical form. In humans, Se effects are mostly through its incorporation into selenoproteins [20] such as antioxidant properties, formation of thyroid hormones, DNA synthesis, the immune response, and fertility [21,22]. Se is an important component of glutathione peroxidases (GPx), thioredoxin reductases (TrxR), and iodothyronine deiodinases (IDD) [23]. Furthermore, Se is involved in the regulation of T lymphocytes, B lymphocytes, NK cells, and neutrophils [24]. It has immune and anti-cancer properties [25] and is essential for testosterone biosynthesis and the formation and normal development of spermatozoa [26]. It also plays a crucial role in the development of the fetus, infant, and child. Some studies showed that Se deficiency in pregnancy can lead to an increased frequency and severity of early and late gestosis, fetal hypotrophy, hypoxia, and increased risk of miscarriage [27].

### 2.2. Symptoms of Selenium Deficiency and Excess

Schwarz and Foltz first showed that Se is an active component of yeast factor 3 and prevents necrotic liver degeneration in rats [28]. In humans, Se deficiency causes several serious diseases, such as Keshan and Keshan–Beck disease, thyroid, cardiovascular or fertility disorders [23,26]. Keshan disease is an endemic cardiomyopathic disorder that mainly affects children and women of childbearing age in some endemic areas of China where the soil is poor in Se. It is manifested by acute heart failure or chronic moderate to severe heart enlargement and can lead to death [29,30,31] but can be prevented by Se supplementation [32]. Kashen–Beck disease is an endemic chronic osteoarthritic disease causing deformity of the affected joints in agricultural regions of eastern Siberia, northern Korea, and central regions of China [33]. This disease usually affects children aged 5–15 and the symptoms include joint pain, morning joint stiffness, impaired elbow flexion and extension, enlarged joints, and limited movement [34,35].

Se deficiency is also related to some thyroid disorders such as the iodine deficiency disorder goitre, cretinism, Hashimoto’s thyroiditis, and Graves’ disease. Se deficiency reduces thyroid hormone synthesis as it decreases the function of selenoproteins, in particular IDD, which are responsible for the conversion of thyroxine (T4) to triiodothyronine (T3) [29,31,36,37,38]. Increased cardiovascular disease mortality has also been associated with Se deficiency [39], probably related to the reflection of sub-optimal GPx4 activity in the prevention of LDL oxidation, with subsequent uptake by endothelial cells and macrophages in arterial blood vessels [40].

Se plays a significant role in the reproductive system [41,42]. A mild Se deficiency can lead to impaired immune function, such as cell oxidation, degeneration, and damage to immune system organs, resulting in reduced immunity and consequently various diseases [4].

Se deficiency in humans is mainly characterised by peripheral myopathy with muscle weakness and pain, cardiomyopathy with an enlarged heart, arrhythmias and chronic congestive heart failure, elevated transaminase and creatine kinase activities, and whitening of nail beds [43]. Other clinical manifestations of Se deficiency in infants and children are growth retardation, alopecia with pseudo-albinism, erythrocyte macrocytosis, and hypothyroidism [43,44,45,46]. Additionally, Se deficiency in children has negative effects on growth and brain development [47,48]. In premature infants, Se supplementation can reduce the incidence of sepsis but further research is still needed [49]. Se plays an important role in fetal development [50,51] and there is an association between lower maternal Se levels and the delivery of small-for-gestational-age children, suggesting that Se deficiency is a possible risk factor for intrauterine growth retardation [52]. Some studies have shown an inverse correlation between Se levels and the risk of preeclampsia, with Se supplementation during pregnancy reducing the incidence of preeclampsia [53]. It is also worth emphasizing that reduced Se levels are also observed in patients with phenylketonuria (due to a diet limiting Se-rich products) [54], cystic fibrosis [55,56], renal failure [57] or autoimmune diseases (increased antioxidant stress) [58].

Se is not an innocuous micronutrient and in excess, is a highly toxic agent [59]. However, globally overt Se toxicity in humans is much less common than deficiency [29,60]. Se toxicity in humans depends on the chemical form, concentration, time of exposure, and several compounding factors [29]. Acute or chronic Se poisoning (selenosis) may lead to gastrointestinal problems, poor dental health, a metallic taste in the mouth, tingling and inflammation of the nose, the typical garlic odor of the breath, diseased nails, and nail loss, dermatitis and skin discolouration, loss of hair fluid in the lungs, pneumonia, lack of mental alertness, peripheral neuropathy, and gastric disorders [29,61,62]. The acute selenium intoxication due to the intake (in the nine cases) of nuts of the *Lecythis ollaria* tree in a Se-rich soil area of Venezuela resulted in vomiting and diarrhoea followed by hair and nail loss and the death of a 2-year-old boy [62].

Population studies in Se-rich soil areas showed elevated urinary Se levels but no definite links to clinical symptoms of selenosis. However, a higher incidence of gastrointestinal problems, poor dental health, diseased nails, and skin discolouration were reported [61]. Other studies conducted in China in the 1960s observed an association between the consumption of plants with a high Se content (grown on soil containing >300 mg kg^−1^ Se) and hair and nail loss, disorders of the nervous system, respiratory system, skin, poor health teeth, garlic breath, and paralysis [63]. Currently, the main risk of Se poisoning may be the uncontrolled intake of Se supplements [64].

Moreover, poor vitamin E status increases Se toxicity and the nutritional need for the element, whereas sulphate counteracts the toxicity of selenate but not of selenite or organic Se and increases Se urinary excretion [29]. Currently, the molecular mechanism of the toxic effect of Se is unclear [65].

### 2.3. Selenium Dietary Requirement in Infants and Preschool Children

Current recommendations for daily Se intake are based on levels maximising the activity of plasma GPx. Based on the EFSA recommended intakes of Se for infants were extrapolated from the estimated Se intake with breast milk of younger exclusively breast-fed infants and taking into account differences in reference body weights. For children, the intakes were extrapolated from the adequate intake (AI) for adults by isometric scaling and application of a growth factor [65]. The estimated Se adequate intake of infants from birth is 12 µg/day and increases with age to 20 µg/day for children aged 4–6. Detailed adequate intake and upper level depending on age is presented in Table 1 [19,65].

The WHO-FAO-IAEA recommended Se intake (World Health Organization–Food and Agricultural Organization of the United Nations–International Atomic Energy Agency) based on epidemiologic evidence derived from areas of China endemic or non-endemic for Keshan disease are presented in Table 2 [66].

## 3. Main Dietary Selenium Sources in Infants and Preschool Children

### 3.1. Selenium in Dietary Products

Se in the diet may occur in organic or inorganic form, with the most common Se compounds in the diet being selenomethionine, selenocysteine, Se-methylselenocysteine, and selenite [67,68,69]. Selenomethionine is the predominant Se species found naturally in foods such as cereals, nuts (especially Brazil nuts), legumes, and yeast. The dietary sources of selenocysteine are of animal origin, especially meat [70], while the sources of Se-methylselenocysteine are vegetables belonging to the *Brassica* (broccoli, Brussels sprout) and *Allium* (garlic and onion) species [71,72]. Selenite occurs naturally in food in small amounts and the main source of inorganic Se compounds in the human diet is supplementation [73]. Inorganic Se is also observed in some foods (e.g., cabbage) and drinking water [74,75,76]. It is worth noting that the Se concentration in food products varies and depends on the origin and culinary processing [77].

### 3.2. Breast Milk and Infant Formula

Human breast milk is the main source of Se for infants but the Se concentration and form varies and depends on the mother’s diet. Se in breast milk is secreted as organic compounds, either in proteins or amino acids [78], and does not contain an inorganic Se form. There are up to nine selenoproteins, mainly glutathione peroxidase (in the largest quantity) and selenocysteine, selenocystine, and selenomethionine, respectively [79]. In European countries, the average Se concentrations in mature breast milk are around 15–20 µg/L [80,81,82,83], which meets the daily AI requirement for this element in infants. However, in some cases, the Se content in breast milk is insufficient [80,81]; therefore, future research should verify whether infants meet the recommended Se intake and assess the influence of the concurrent diet of lactating mothers on the Se content of their milk, especially in mothers on a diet poor in Se-rich products. Table 3 presents Se content in human milk and infant formula.

Formula is the basis of nutrition for infants who cannot be breastfed. These are special preparations that are supposed to resemble breast milk as closely as possible. The Se content in formula for infants is varied. According to the Delegated Regulation (EU), the Se level is between 3.0 and 8.6 µg per 100 kcal regardless of the type of formula (infant formula, follow-on formula, and formula for special medical purposes) [84,85]. At the same time, the United States Food and Drug Administration (FDA) recommends levels between 2.0 and 7.0 μg per 100 kcal [86]. The EFSA Panel on Nutrition notes that exclusively formula-fed infants being fed formula with the currently highest permitted concentration of selenium as 8.6 μg per 100 kcal will consume an average of 43 μg/day Se using mean formula intakes in the first half year of life of 500 kcal/day as a basis. However, infants with increased nutritional needs consuming the maximum energy of about 700 kcal/day would have a Se intake of 60 μg/day. Therefore, infants aged 4–6 months and 7–11 months who are predominantly fed formula with the highest Se concentration may exceed the UL of 45 µg/day and 55 µg/day, respectively, when such milk is their only source of nutrition [19]. There is a need for studies evaluating the Se intake of infants exclusively formula-fed.

Only the inorganic form of Se is currently permitted in infant formula milk in most countries, except for Australia and New Zealand, which allow the use of selenomethionine [87]. In the European Union, the possible Se forms occurring in infant formula and follow-on formula are sodium selenate and sodium selenite. Additionally, food for special medical purposes includes sodium hydrogen selenite or Se-enriched yeast [88]. The FDA does not recommend a particular selenium form for infant formula [78].

**Table 3 nutrients-15-04668-t003:** Se content in breast milk and infant formula.

Food Source	Average Content	Comments
Breast milk	2.2–3.0 μg/100 kcal *	Se content depends on the maternal diet
Infant formula	3.0–8.6 μg/100 kcal	According to the Delegated Regulation (EU)
	2.0–7.0 μg/100 kcal	According to the FDA

* Taking into account that 100 mL of breast milk has 67 kcal [89].

### 3.3. Main Dietary Selenium Sources after Weaning in Children 

After weaning, the main Se sources besides human milk and formula are fish, meat, eggs, dairy products, and cereals (Table 4) [90]. Consumption of these food products should fully cover the demand for this element.

#### 3.3.1. Fish Products

Fish are a very good source of Se in the human diet but the content depends on the species, place of catch, and method of culinary processing [95,96]. Marine fish have a higher Se content compared to freshwater fish [95]. Examples of marine fish include tuna, bluefish, red snapper, and sardine, with freshwater fish including pike (northern), bass (largemouth), and walleye [75,97,98]. Additionally, Se concentration in fish depends on the origin, e.g., Se concentrations in tuna from Spain and Portugal are 0.92 ± 0.01 µg/g [98], from New Jersey is 0.43 ± 0.04 µg/g [99], and from Japan is 0.75 µg/g [100]. It is worth noting marine fish accumulate relatively high levels of Se which are strongly correlated with the mercury content in several organs, including muscle tissue [97,101]. However, some studies showed that Se counteracts mercury, especially methylmercury toxicity [101,102]. The mechanism of the observed relationship between mercury and selenium is not fully known.

The Se content in fish from different locations varies greatly, ranging between 60–630 ng/g, with a high Se content observed in fish from New Zealand and Australia [96]. The fish preparation process affects the Se, with cooking in water reducing Se by 36–46% [103] but mussels retrain Se when boiled or fried [95].

Food safety in children’s nutrition should be taken into account, as the risk of heavy metals and other chemical and biological hazards will be of key importance in fish consumption. Therefore, fish from a verified source and after prior heat treatment (reducing the pathogenic microorganism) should only be served [104]. Barone et al. suggested that children should consume fish in moderation because a large consumption pattern, especially of swordfish and tuna, might be of health concern regarding the mercury content. However, all analysed fish mercury levels were not above the European Community regulatory limits in their study [105].

According to the European Society for Pediatric Gastroenterology Hepatology and Nutrition (ESPGHAN) recommendations, fatty marine fish are generally recommended for infants after weaning, such as Atlantic herring, farmed Norwegian salmon, sprat, sardines, farmed trout, flounder, cod, Atlantic mackerel, and hake. Infants should not be given predatory fish such as swordfish, shark, king mackerel, tuna, and tilefish. Fish should be given in small portions, no more than 1–2 times a week, observing the child’s reaction (e.g., any allergic reactions) [106].

#### 3.3.2. Meat, Eggs and Milk Products

Meat, eggs, and milk products are rich in Se, although its amount is lower than in fish. However, due to the high consumption of these products in many countries, they are an equally valuable source of this element. The average Se content in these products is presented in Table 4.

The Se concentration in meat varies, depending on the Se content in the food consumed by animals [107]. It also depends on the type of meat, being higher in pork (0.14–0.45 µg/g) than in lamb (0.03 µg/g) or beef (0.08–0.2 µg/g) [108]. A high Se content is also observed in offal (0.17–0.30 µg/g) [109].

The Se content in eggs is influenced by the Se level in the animal diet [110]. The average Se content in a whole egg is about 0.17 ug/g, which means that eating one egg (55 g) will allow the consumption of 9.35 µg of Se [94]. Some studies showed that eggs from hens fed natural Se contained more Se than those from hens fed selenite [111]. Other studies evidenced that hens fed barley had a significantly higher concentration of Se in the egg yolk and egg white in comparison with those receiving sodium selenite [112]. Hens fed from organic sources of Se produce eggs containing 10 to 29 μg of Se [113].

The Se content in milk and dairy products is much lower (Table 4) but since they are a natural source of calcium, their consumption in humans should be increased. According to Yanardag et al., Se content in dairy products varies and depends on origin: in butter (0.03–0.22 µg/g), coffee cream (0.03–0.25 µg/g), cheeses (0.02–0.29 µg/g), and milk powder (0.06–0.10 µg/g). Additionally, they observed that Se-rich sources were proteinaceous foods such as white cheese [114]. Moreover, the Se content in milk varies widely depending on the season, with higher Se levels in summer than in winter; however, this difference does not translate into increased consumption of Se [96].

Regarding children’s nutrition, meat, eggs, milk and dairy products are valuable sources of nutrients for the developing body, so they should be part of the daily diet, which makes them an important source of Se in the diet.

#### 3.3.3. Frutis and Vegetables

Fruits and vegetables contain low concentrations of Se but some are a valuable source of Se-methylselenocysteine with important health benefits, including vegetables from the *Brassica* and *Allium* species.

*Brassica* species can accumulate Se because they can replace Se in proteins but this accumulation is limited by the Se concentration in the soil [93,115]. It has been shown that *Brassicaceae* family can accumulate up to 300-fold more Se in their tissues when grown in Se-rich soil. In Gui et al.’s study was evidenced Se contents in Broccoli florets were significantly higher under Se yeast treatments (300-fold) and under selenite treatments (50 fold) than under non-selenized treatments [116]. The Se concentrations in broccoli, brussels sprouts, and cabbage grown in non-Se-enriched soil are 0.01–0.03 µg/g, 0.004–0.06 µg/g, and 0.001–0.02 µg/g, respectively [117]. Additionally, these vegetables, apart from selenium compounds, i.e., Se-methylselenocysteine and seleno glucosinolates, are rich in flavonoids [118].

*Allium* species can also accumulate Se through the metabolic absorption of sulphur [119]. The average Se content in garlic is 0.15 µg/g and onion is 0.12 µg/g [120,121]. However, the enrichment of crops causes the Se content in allium species to be much higher and amounts to 68–1355 µg/g and 96–601 µg/g (in dry mass), respectively [69,122,123]. Additionally, *Allium* species are rich in various active compounds, especially organosulfur and polyphenols [124,125]. Allium vegetables are a good source of selenium, although due to limited consumption by young children, it may not significantly affect the selenium content in their diet.

The Se content of other vegetables and fruits is low and does not increase the Se intake in the diet. However, they are a valuable source of antioxidants and vitamins necessary for proper development.

#### 3.3.4. Brazil Nut

Brazil nuts are the one of richest known food sources of Se and are obtained from the *Bertholletia excelsa* tree from South America [126,127]. The average Se content ranges from 2 to 20 µg/g [128]. However, the content in individual nuts varies significantly (0.03–512 µg/g) [129] depending on the Se content and bioavailability in the soil [130,131]. The Se concentration of Brazil Nuts from the Brazilian Amazon basin varies from in the Mato Grosso state (2.4 µg/g) and in the Acre state (3.0 µg/g) to in the Amazonas state (66.1 µg/g) and Amapa state (51.2 µg/g) [131]. Additionally, Brazil nuts are a good source of nutrients including protein, fibre, minerals, and vitamins, and therefore have a variety of potential health benefits [132,133]. However, due to the possible high Se content, children should not consume Brazil nuts in excess (1 Brazil nut contains approximately 50 µg) [69]. It should also be emphasized that even though Brazil nuts are a good source of Se, care should be taken to avoid the possible toxic effects associated with a chronically high radium (Ra) and barium (Ba) intake. Of note is that the toxic values of Ba (sub-acute exposure to Ba can cause muscle weakness) [134,135] and Ra taken with food are not clearly defined but Ba toxicity has been reported with ingestions as small as 200 mg Ba/kg/day [136,137].

#### 3.3.5. Cereals and Yeast

Cereal products are not particularly rich in Se (0.01–0.55 µg/g) but they can increase the dietary Se content due to their daily consumption [92]. The difference in the Se content of whole wheat (mean = 0.13 µg/g) and white bread (mean = 0.09 µg/g) is not statistically significant, although whole wheat bakery products are much richer in Se [96]. 

Regarding gluten-free flour, the richest source is amaranth (0.5 μg/g), followed by native buckwheat flour (0.41 μg/g) and corn flour (0.46 μg/g), with the lowest Se content in rice flour (0.14 μg/g). Interestingly, Adams et al. found no long-term changes in the distribution of Se concentrations in wheat grains over 17 years in regions around the United Kingdom [138].

Fertilisers are useful to increase the cereal Se content. This beneficial effect was observed in the Finnish national Se supplementation programme, where the Se content increased on average 15 fold [139,140]. 

The Se content in bakery products can also be increased by adding yeast, as the commercial dried product contains 2.5 mg Se/g (range 1–2.4 mg Se/g); however, yeast consumption is low [141]. The addition of Se-rich yeast to bakery products could significantly improve future Se intake [142].

## 4. Bioavailability of Selenium

Se bioavailability is dependent on many factors, of which the main factor is the chemical form of this element. As previously mentioned, organic compounds are more bioavailable than inorganic (selenomethionine and selenocysteine > 90%, selenate and selenite > 50% absorbed) [143,144]. Additionally, it can be affected by some other components of the food matrix [145,146,147]. The major food ingredients affecting bioavailability are carbohydrates, proteins, fat and dietary fibre, and minor food components including vitamins, toxic metals, and oligoelements [148].

The carbohydrate content in fish, shellfish, and seaweed increased Se bioavailability [149] because the carbohydrates form micelles which can enhance the partition of hydrophobic molecules in the aqueous solution [148,150]. Se bioavailability decreases with increasing protein content in fish, shellfish, and seaweed samples [151].

Some studies in animal models have shown increased Se bioavailability with an increase in the proportion of polyunsaturated fatty acids in the diet [152]. However, other studies showed that the resulting differences in fatty acid composition in the human diet, with a higher content of polyunsaturated fat acids, do not influence Se absorption [153]. In vitro studies suggest that the fat content has no effect on Se bioavailability in fish and shellfish [151]. Shen et al. showed that removing fat from milk significantly increased Se bioavailability, possibly due to the better protein digestibility of skimmed milk [154].

Se is most easily absorbed in the presence of vitamins A, D, and E. They increase Se bioavailability, while heavy metals (especially mercury) and fibre decrease it [155,156]. Dietary sulphur (especially from methionine) may compete with Se for absorption. Table 5 summarises Se bioavailability depending on food ingredients.

Additionally, parameters related to the human body, i.e., Se status, age, sex, or lifestyle factors may influence on bioavailability of Se [143].

## 5. Selenium Intake in Elimination Diets

In healthy, normally developing children, the Se concentration will depend primarily on the diet. However, in children in whom it is necessary to introduce certain dietary restrictions related to a disease (e.g., phenylketonuria, food allergies), meeting the demand for Se may be insufficient.

Phenylketonuria diets are low in protein products, with no Se-rich products such as fish, meat, eggs, and whole grains. Therefore, phenylketonuria patients often have low serum Se levels. In Okano et al.’s study, the serum Se concentration in all analysed patients with PKU (*n* = 11) in the ages of 4–38 years was lower than the reference value and amounted to 56.6 ± 21.2 µg/L [157]. The Se concentration and GPx activity in the plasma and erythrocytes of 87 patients participating in the German collaborative study of phenylketonuria (mean age 9.7 years) were negatively correlated with the quality of dietary management (mean plasma phenylalanine value). However, despite the low Se levels, the children did not show any clinical symptoms of deficiency [54]. Another study assessed 54 children aged 4 to 10 years with phenylketonuria before and after the use of a mixture of amino acids with Se. Se supplementation through protein preparations for phenylketonuria effectively improved the nutritional status of Se [158]. Moreover, according to a Chanoine study, in patients with isolated Se deficiency (e.g., in patients with phenylketonuria on a low-protein diet), the metabolism of peripheral thyroid hormones was disturbed, but no changes in thyrotropin concentration or clinical symptoms of hypothyroidism were observed, which suggests that these patients are euthyroid. Although this statement may be questionable because euthyroidism should be defined on the basis of FT4/FT3 and other clinical parameters, it is suggested that Se supplementation is recommended for patients following a diet low in natural protein. Se supplementation may be indicated to optimise GPx activity in tissues and prevent potential damage caused by oxidative stress [46]. Therefore, Se supplementation is recommended in patients following a diet low in natural protein [159,160].

Recently, attention has been paid to the occurrence of Se deficiency in food allergic children (IgE mediated). This may be related to both an appropriate elimination diet (allergies to fish, cow’s milk protein, etc.) as well as an increased immune response [161,162,163,164].

In animal model studies, oral Se supplementation may modulate allergic reactions to cow’s milk protein by reducing specific antibody responses [164]. Kamer et al. reported that children with allergies had decreased Se concentrations and GPx, which increased after an elimination diet, suggesting that Se may play a role in the pathogenesis of food allergies. Additionally, this observation indicates the need to monitor trace elements content in the diet of children with food allergies [163].

## 6. Selenium Dietary Intake in Infant and Preschool Children—Overview of Available Studies

There are few publications describing the Se intake of preschool children. Up to now, there are three studies assessing Se intake in preschool children [18,165,166] and three studies in infants [165,167,168] (Table 6). This may be due to many factors, such as Se is often omitted in studies on dietary intake which focus mainly on macronutrients and elements that have long been associated with problems with children’s development, e.g., calcium or iron. Obtaining a 24 h food questionnaire may also be problematic, as this requires a lot of parental involvement and the amounts consumed by the child may be difficult to determine.

Depending on the region, the problem of Se intake is varied. In Brazil, children consumed so much Se that the amounts were considered potentially toxic, whereas the intake was insufficient in the Philippines and Malawi [18,165,168]. Improving intake to adequate could be achieved by adding one additional serving of a product such as eggs or milk to the diet [165,166,167]. This proves that providing children with the basic amounts of recommended food products can easily prevent deficiencies.

Research also shows that some infant milk formulas do not contain the right amounts of Se to ensure adequate daily intake; however, these studies are from 1982 [167]. Currently, the Se content in infant milk formula is strictly defined by the Delegated Regulation (EU) and FDA [84,85,86].

## 7. Serum Selenium Concentration in Infant and Preschool Children—Overview of Available Studies

Table 7 presents thirteen studies examining the serum Se concentrations in infant and preschool children. The Se concentration in serum varied according to area, with Se deficiency in children from Malawi and a Se excess in children from Brazil [18,168,169]. This implies that the food consumed on a daily basis in a given region has a significant impact on the children’s Se status. Serum Se levels could be increased by the biofortification of a widely consumed food product [170].

For comparison, age-specific reference intervals based on the 2.5 and 97.5 percentiles of data derived from a healthy pediatric population for Se concentrations in serum are presented in Table 8 [1].

There have also been several results describing Se concentrations in newborns from birth through the first period of life, showing that the Se concentration is low and varies depending on the months of life [18,168,171,172,173]. In most of the analysed studies, mean Se concentrations in serum were low, but well within the reference intervals shown in Table 8. Only Zyambo et al.’s study had lower values, but it was within a group of children with severe acute malnutrition [170].

It is worth noting that most of the deficiencies can be easily modified by administering supplements enterally. However, it is important to select the appropriate dose and form, which affects its bioavailability. Nonetheless, the most effective way to maintain the correct level of Se in children is breastfeeding [174,175].

**Table 7 nutrients-15-04668-t007:** Summary of available studies examining the serum Se concentrations in infant and preschool children.

Study	Country	Patients	Observations
Zyambo et al., 2022 [170]	Zambia	269 children: with severe acute malnutrition (n = 19), with stunting (n = 164), and children without stunting (n = 86), received Se supplementation Age: 15 (4–23) months Sex: both	The median Se levels were 32.37 μg/L (21.32–62.38) among unstunted children, 45.01 μg/L (41.85–64.74) among stunted children, and 71.85 μg/L (47.37–89.22) among severely malnourished children Se deficiency is widespread in Lusaka province and could in part be related to socio-economic status, therefore supplementation or agronomic biofortification is needed
Martens et al., 2015 [18]	Brazil	129 children: 41 receiving 15–30 g of brazil nuts 3d/w and 88 that did not receive nuts age: 4.7 ± 0.9 (3.1–6.3) years old in the study group; 4.5 ± 1.2 (2.1–6.6) in control sex: both	Plasma Se concentration in supplemented children: 107.29 ± 27.15 (73–172) μg/L Plasma Se concentration in controls: 83.56 ± 23.32 (47–142) μg/L Plasma Se levels of supplemented children were significantly higher than in the control group but the Se concentration was higher than the accepted cutoff (>84–100 μg/L) for both groups Se supplementation of Brazilian children is not necessary and could even lead to Se poisoning
Darlow et al., 1995 [49]	New Zealand	15 children age: 0–5 days sex: both	Plasma Se concentration: 39.69 ± 3.16 μg/L After 1 month of feeding with standard formula: 21.32 ± 0.79 μg/L After 3 months of feeding with standard formula: 31.58 ± 3.16 μg/L There was a significant drop in plasma Se levels between birth and the first month of life but compared to a group of infants supplemented with Se 17 μg/L to resemble breast milk composition, there were no differences in growth parameters or thyroid function
Perez-Plazola et al., 2023 [169]	Malawi	387 children age: 7 (1.2) months sex: both	Plasma Se at baseline was 47.41 (28.48) μg/L and 50.59 (28.58) μg/L The children had inadequate plasma Se concentrations given the minimal cutoff of 70 μg/L for optimal body functioning The provision of one additional egg per day for 6 months did not increase plasma Se levels, which is thought to happen because of high rates of stunting and underweight status in those children
Flax et al., 2014 [168]	Malawi	526 children of HIV-infected mothers age: 2–6 weeks old sex: both	Plasma Se: 55.6 ± 16.3 μg/L at 2–6 weeks and 61.0 ± 15.4 μg/L at 24 weeks The children did not have adequate plasma Se levels at any time The Se concentrations correlated with baseline tertile, being the lowest (40.1 ± 9.3) for low tertile, medium (57.0 ± 3.9) for middle tertile, and the highest (73.9 ± 7.5) for high tertile Maternal plasma Se levels and breast milk Se concentrations correlated with infant plasma Se
Olmez et al., 2004 [176]	Turkey	131 children: 88 with acute gastroenteritis, 43 healthy age: 2–24 months sex: both	Control group: 74.36 ± 10.65 μg/L Study group: 62.41 ± 13.06 μg/L on admission and 81.73 ± 17.10 μg/L 7–10 days after the end of symptoms Children suffering from gastroenteritis had significantly lower plasma Se levels on admission than healthy children but higher after the end of symptoms Se levels did not correlate with the severity of the symptoms
Gibson et al., 2011 [177]	Zambia	476 children age: 6 months sex: both	Baseline serum Se concentrations: 48.95 (47.37, 50.53) μg/L, and 49.75 (47.37, 52.11) μg/L Plasma Se levels were defined as below adequate for maximal activity of plasma GPx and selenoprotein P (~78.96–94.75 μg/L) Children with higher baseline plasma Se showed a better response to Se fortification, probably because of the decreased expression of selenoproteins in children with low Baseline plasma Se
Daniles et al., 1996 [174]	Australia	38 preterm infants fed parenterally with or without Se supplementation age: <7 days sex: both	Baseline plasma Se: 28(3) μg/L and 27(3) μg/L Baseline plasma Se levels were inadequate Se supplementation with 3 µg/kg/day prevented health deterioration but did not achieve levels similar to breastfed infants
Christodoulides et al., 2011 [178]	United Kingdom	44 children with intractable epilepsy divided into 2 age groups (2–3 and 4–6) age: 2–6 years sex: both	Baseline mean plasma Se levels: 2–3-year-old group 75.80 (15.00) μg/L 4–6-year-old group 77.38 (20.53) μg/L (further divided into male and female groups with concentrations of 64.74 (45.80–89.22) μg/L and 63.96 (45.01–82.90) μg/L, respectively) Plasma Se levels were within the range of GOSH reference (39.48–102.65 μg/L for 2–4-year-olds and 55.28–134.23 μg/L for 4–17-year-olds) The children were either on a classical ketogenic diet or MCT ketogenic (the main fat source is MCT fatty acids) and there were no significant differences in plasma Se levels between these two groups
Bogye et al., 1998 [171]	Hungary	36 preterm infants (mean gestational age 27 weeks) age: one day sex: both	Baseline serum concentration in the control group: 34.4 (20.4) μg/L Serum concentration after 14 days: 26.1 (16.6) μg/L Baseline serum concentration in the study group: 36.1 (12.8) μg/L Serum concentration after 14 days of supplementation (4.8 mg yeast–Se containing 5 μg Se daily with nasogastric drip): 43.5 (7.9) μg/L Se concentration in the control group decreased during the first 14 days of life and increased significantly in the supplemented study group There were no side effects and the intervention was considered safe
Linday et al., 2002 [179]	USA	39 children undergoing placement of tympanostomy tube for frequent ear infections and/or persistent middle ear effusion	Serum Se concentration: 110.54 ± 16.58 μg/L There was no difference in plasma Se concentrations from published values Cod liver oil and multivitamins containing Se were proposed as antioxidants to prevent free radical-induced lipid peroxidation that could lead to otitis media The intervention improved the time of antibiotic therapy and symptom re-occurrence
Strauss et al., 2010 [175]	USA	15 children with maple syrup urine disease age: 0–36 months sex: both	Baseline Se concentration: 56.7 (10.9) μg/L Baseline plasma Se in studied children was inadequate Supplementation of 4–9 μg/kg/day raised plasma Se levels, but they remained lower than in the healthy population (83.4 (11.0) μg/L vs. 110–160 μg/L normal)
Li et al., 1999 [172]	Austria, Slovenia	25 neonates—5 Austrian, 20 Slovenian age: 0 (umbilical cord blood at the time of delivery) sex: both	Austrian neonates: 42 ± 6 μg/L Slovenian neonates: 34 ± 7 μg/L All infants had plasma Se levels lower than their mothers, which is consistent with previous findings

**Table 8 nutrients-15-04668-t008:** Reference intervals (2.5 and 97.5 percentiles) from a healthy pediatric population for serum Se concentrations [1].

Age	Serum Se Concentrations (µg/L)
<1 month	15–107
1–2 months	15–100
2–4 months	10–93
4–12 months	13–116
1–5 years	34–129

## 8. Concluding Remarks—Nutritional Recommendation for Infants and Preschool Children Regarding Selenium Intake

The current literature review indicates a significant problem with meeting the Se needs of preschool children and infants [18,157,165,168,170,174,177]. Table 9 summarises the dietary recommendations that should contribute to improving the serum Se concentration in this age group.

## 9. Summary

The main focus of this article is to provide an all-encompassing account of the significance of selenium in the diet of young children (infants and preschoolers). It is important to note that the existing literature on selenium can be categorized into two distinct groups: (1) descriptions of dietary sources, bioavailability, physiological functions, and deficiency symptoms, and (2) cross-sectional examinations of selenium levels in particular populations. Our research takes a comprehensive approach to this issue. The population under study, namely children aged up to 5–6 years, is of particular significance due to two seemingly opposing reasons: these children are at higher risk of selenium deficiency owing to food allergies or selective feeding, while, conversely, they may also suffer from overdosage due to smaller requirements and food fortification. To the best of our knowledge, this is the first piece of research to concentrate specifically on the difficulties encountered by this group. However, this study’s scope is constrained by the absence of a meta-analysis, which precludes statistical interpretation of the aggregated results.

Currently, there is a lack of epidemiological studies regarding serum selenium concentration, especially compared with estimated dietary intake of selenium in infants and preschool children. Therefore, there is a lack of nutritional strategies for infants and young children regarding selenium intake.

Selenium deficiency is very common in areas with low selenium soil, including highly developed countries in most of Europe. It may contribute to a developmental disorder, hypothyroidism, and lowering the immune response. Therefore, screening of the Se content in infants and children in the context of possible deficiencies may be necessary. This will be of particular importance in children with an increased risk of deficiency, i.e., patients with phenylketonuria or food allergies. However, due to the awareness of possible selenium deficiency, it is worth monitoring the levels in patients with symptoms of deficiency, e.g., thyroid problems.

## 10. Conclusions

Due to possible Se deficiencies in infants and preschool children, epidemiological studies are needed to determine Se intake and serum Se concentrations in these population groups.

## Figures and Tables

**Figure 1 nutrients-15-04668-f001:**
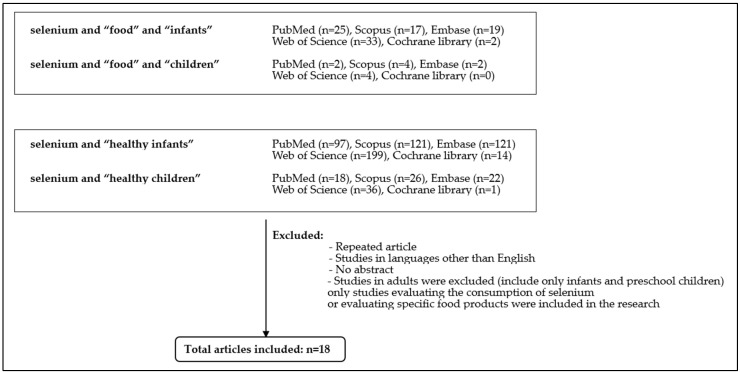
Search strategy.

**Table 1 nutrients-15-04668-t001:** Recommended adequate intake and upper level of Se for infants and preschool children.

Age	AI µg/Day	UL µg/Day
0–6 months	12	45 *
7–11 months	15	55
1–3 years	15	70
4–6 years	20	95

* According to the EFSA, the upper level (UL) was extrapolated for infants from 4–6 months when there was no data for younger infants.

**Table 2 nutrients-15-04668-t002:** The WHO-FAO-IAEA recommended Se intakes (µg/day).

Age Group	Assumed Weight (kg)	Average Normative Requirement of Se (µg/Day)		Recommended Nutrient Intake (RNI) of Se
		Per kg Body Weight/Day	Total/Day	
0–6 months	6	0.85	5.1	6
7–12 months	9	0.91	8.2	10
1–3 years	12	1.13	13.6	17
4–6 years	19	0.92	17.5	22

**Table 4 nutrients-15-04668-t004:** Main Se sources in children aged 1 year and older.

Food Source	Average Content µg/g	Average Se Content µg/per Serving (Serving Size)
Fish [75]	0.4–4.3	20–215 µg [50 g]
Meats (mussels) [75]	0.03–0.45	1.5–22.5 µg [50 g]
Yolk from egg [91]	0.12–0.42	1.2–4.2 µg [10 g–½ piece]
Cereals [92]	0.01–0.55	0.75–41.25 µg [75 g]
Broccoli [93]	0.02	1 µg [50 g]
Cow’s milk [94]	0.01–0.02	0.5–1 µg [50 mL]
Gouda cheese [94]	0.08	1.2 µg [15 g–1 one slice]
Yoghurt [94]	0.02	1 µg [50 g]

**Table 5 nutrients-15-04668-t005:** Bioavailability of Se depending on food ingredients.

Food Ingredients	Effect on Bioavailability
Proteins	↓ Se bioavailability with increasing protein content in fish and seafood
Fats	↑ Se bioavailability with increase polyunsaturated fatty acid in diet—animal model study ↑ Se bioavailability with reducing the fat content in milk
Carbohydrates	↑ Se bioavailability—analysed in fish and seafood
Dietary fibre	↓ Se bioavailability
Vitamins A, D, and E	↑ Se bioavailability
Sulphur	↓ Se bioavailability sulphur in diet may compete with Se for absorption

**Table 6 nutrients-15-04668-t006:** Summary of available studies examining the Se dietary intake in infant and preschool children.

Study	Type of Study	Country	Patients	Observations
Martens et al., 2015 [18]	interventional	Brazil	129 children: 41 received 15–30 g of Brazil nuts 3 day/w and 88 did not receive nuts age: 4.7 ± 0.9 (3.1–6.3) years old in study group; 4.5 ± 1.2 (2.1–6.6) in control sex: both	Median (range) Se intake in the supplemented group: 155.30 (98.7–195.3) µg/day Median (range) Se intake in the control group: 44.40 (33.9–53.20) µg/day Given the EAR for Se of 17 µg/day for 1–3-year-olds and 23 µg/day for 4–8-year-olds, the children consumed excess Se even without supplementation with nuts Se intake was considered highly probable toxic in the group that received nuts
Mak et al., 2020 [165]	interventional	the Philippines	A total of 2642 children in given age groups: 792 children age: 1–2 years sex: both 1136 children age: 3–4 years sex: both 714 children age: 5 years sex: both	mean Se daily intake: 30.2 µg (23% of children had inadequately low Se intake) after supplementing 1 serving (180 g) of powdered milk: 31.2 µg/day after supplementing 1 serving (180 g) of YCM (young children’s milk formula): 35.5 µg/day mean Se daily intake: 45.9 µg (8% of children had inadequately low Se intake) after supplementing 1 serving (180 g) of powdered milk: 46.9 µg/day after supplementing 1 serving (180 g) of PCM (preschool children’s milk): 50.8 µg/day mean Se daily intake: 51.7 µg (4% of children had inadequately low Se intake) after supplementing 1 serving (180 g) of powdered milk: 52.8 µg/day after supplementing 1 serving (180 g) of PCM (preschool children’s milk): 57.1 µg/day In all groups, adding a portion of YCM/PCM significantly improved plasma Se levels; therefore, meeting the guidelines for daily diary recommendations can significantly decrease the number of children with inadequately low Se intake
Caswell et al., 2021 [166]	interventional	Malawi	660 children (8% underweight, 1% wasted) age: 7.4 ± 1.2 months sex: both	Mean (SD) estimated usual intake of Se 19.7 ± 0.2 µg/day Children received 1 additional egg per day Baseline intake was adequate in most children (1% prevalence of inadequacy) but the intervention reduced the prevalence of inadequacy and improved the intake of fat, protein, vitamin A, riboflavin, vitamin B12, and choline
Smith et al., 1982 [167]	observational	USA	28 children age: 3 months sex: both	group I (n = 8) received human milk only group II (n = 20) received formula (Enfamil) mean Se intake in group I: 10.08 ± 2.96 µg/day mean Se intake in group II: 7.22 ± 1.26 µg/day Formula-fed infants had significantly lower daily Se intake than breastfed and did not meet the recommendation of the National Research Council of 10–14 µg Se/day Correlation between intake and plasma levels showed that not only the amount of Se in formula but also its bioavailability should be taken into consideration
Flax et al., 2014 [168]	randomized controlled trial	Malawi	526 children of HIV-infected mothers age: 2–6 weeks old sex: both	mean Se intake at 2nd–6th week of life: 10.1 ± 8.2 µg/day mean Se intake at 24th week of life: 7.7 ± 5.7 µg/day According to the WHO-recommended Se intake, 39% of infants did not achieve sufficient Se intake at 2–6 weeks or 24 weeks postpartum. The mean BMI of mothers was <23 kg/m^2^

**Table 9 nutrients-15-04668-t009:** Dietary Se recommendations for infants and young children.

Infants
Breastfeeding is recommended until at least 6 months; however, the diet of breastfeeding women should include products rich in Se. Nonetheless, no studies confirming the correlation between the Se concentration in the mother’s diet and the Se content in the milk have been published [27,79,80,81,180];
In the case of infants who cannot be breastfed, it is recommended to use a milk formula (Se content regulated by the EU and FDA). However, in infants with increased energy demand and the use of milk with the highest permitted Se concentration (8.6 µg/100 kcal) and feeding exclusively with this milk (in infants over 4 months), the UL value may be exceeded [19].
**Expanding Diet**
Introduce Se-rich products into the child’s diet: meat, fish, eggs, cereals, *Brassica* vegetables; For children with allergies, products with a high Se content, i.e., fish and meat should be replaced with other Se-rich products (eggs, cereals, Brassica vegetables) or consider supplementation; It is recommended to monitor serum Se and introduce Se supplementation in children with phenylketonuria [54,158,163].
**Dietary Recommendation from 1 to 3 Years Old**
The diet should be varied and rich in products containing Se; Children should not consume Brazil nuts in excess (1 Brazil nut has around 50 µg) *; Se-containing supplements in toddlers and children should be used with caution, based on individual needs [18,19,69].

* The average Se content in Brazil nuts is assumed to be 10 µg/g, and one nut has an average of 5 g [69].

## Data Availability

No new data were created or analysed in this study. Data sharing is not applicable to this article.

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
