# Peer review of "Selenium in Infants and Preschool Children Nutrition: A Literature Review"

_nutrients, 2023, doi:10.3390/nu15214668_

Round 1
Reviewer 1 Report
Comments and Suggestions for Authors
An interesting review of the literature on dietary selenium intake in infants and preschool children. The above topic is worth attention because selenium is an element necessary during the period of intensive growth of the body and both its deficiency and excess may be dangerous. Additionally, the results of research on this subject may vary, depending on the place its conducted and the selenium content in the environment.
The manuscript requires minor revisions before it can be published:
Affiliation no. 2 (line 7) should be removed if the paper is from one department.
The keyword - microelement is too general, it is better to replace it with specific - selenium. Similarly, the word "diet" seems unnecessary, because next is “infant nutrition” and “preschool children nutrition”
In the line 32 should be “Poland” instead “Polish”. Please carefully check English.
A literature search scheme would be useful in the methodology - how many papers were found? How many papers were taken into account to detailed review? etc.
In the line 61 should be: glutathione peroxidases.
Please carefully check references, no. 28 is not Schwarz and Foltz (line 70).
Please correct the brackets on line 76.
Instead “immune organs” it should be: “immune system organs”.
To make the manuscript more relevant to the title, please take in consideration removing some information, for example point 2.2. concerns the general effect of selenium on the body, only a small part applies to children.
Similarly, in point 4 regarding the bioavailability of selenium, mushrooms are rather not recommended in the diet of young children.
In the line 176 should be: “breast milk is insufficient …”.
The sentence: “patients often have low serum Se levels. content.” should be improved (line 342-343).
The units in table 5 should be standardized: “µg/d” or “µg/day” or “per day …. µg” or “µg Se/day”; “mcg/L” or “µg/L” or “µg Se/L”.
“µg/l” should be corrected.
For a better comparison, it should be considering converting micromoles and also µg/dl.
In the same table please check in Caswell et al. should be “mg”??
Similarly, in Daniles et al. 3 g/kg/day is correct??
Comments on the Quality of English Language
Please carefully check English.
Author Response
Response to the comments made by the reviewers
Manuscript ID: nutrients-2666192
Title: Selenium in infants and preschool children nutrition, threat of excess or deficiency? A critical review
We would like to thank the Reviewers for their careful review of our manuscript and for providing us with some suggestions to improve its quality. We have carried out a major revision of the manuscript, and we believe the paper has improved significantly.
According to the Reviewers' suggestion, the manuscript has been carefully checked and corrected. The changes in the manuscript have been highlighted in red.
Below we sequentially address all of the points raised by the Reviewers.
Reviewer 1:
Firstly, we would like to express our profound thanks to the Reviewer for devoting time to reviewing our manuscript, the corrections and suggestions. We have carried out a major revision of the manuscript, and we believe the paper has improved significantly.
The Reviewer's comment: Affiliation no. 2 (line 7) should be removed if the paper is from one department.
The authors' answer: We have removed this information.
The Reviewer's comment: The keyword - microelement is too general, it is better to replace it with specific - selenium. Similarly, the word "diet" seems unnecessary, because next is “infant nutrition” and “preschool children nutrition”
The authors' answer: According to the Reviewer's suggestion, we have modified keyword.
The Reviewer's comment: In the line 32 should be “Poland” instead “Polish”. Please carefully check English.
The authors' answer: We have corrected it. According to the Reviewer's suggestion, we checked English.
The Reviewer's comment: A literature search scheme would be useful in the methodology - how many papers were found? How many papers were taken into account to detailed review? etc.
The authors' answer: According to the Reviewer's suggestion, we have added search scheme.
Figure 1. Search strategy.
The Reviewer's comment: In the line 61 should be: glutathione peroxidases.
The authors' answer: We have corrected it.
The Reviewer's comment: Please carefully check references, no. 28 is not Schwarz and Foltz (line 70).
The authors' answer: It was an editorial mistake. We have corrected this reference.
The Reviewer's comment: Please correct the brackets on line 76.
The authors' answer: We have corrected it.
The Reviewer's comment: Instead “immune organs” it should be: “immune system organs”.
The authors' answer: We have corrected it.
The Reviewer's comment: To make the manuscript more relevant to the title, please take in consideration removing some information, for example point 2.2. concerns the general effect of selenium on the body, only a small part applies to children.
The authors' answer: According to the Reviewer's suggestion, we have removed some information that did not concern children.
The potential causes of Kashin-Beck disease are mycotoxins present in grain, trace mineral deficiencies in diet (including Se) and a high level of fulvic acid in drinking water [35]….
Se plays a significant role in the reproductive system [41]. Se deficiency leads to impaired spermatogenesis in men through reduced production of phospholipid per-oxide glutathione peroxidase (PHGPx) and selenoprotein P [41]. Se supplementation can improve reproductive efficiency in women and prevent pregnancy complications such as preeclampsia [42].
The Reviewer's comment: Similarly, in point 4 regarding the bioavailability of selenium, mushrooms are rather not recommended in the diet of young children.
The authors' answer: According to the Reviewer's suggestion, we have removed some information that did not concern children.
In contrast, Se-containing polysaccharides in mushrooms characterise the low Se bi-oavailability [143].
The Reviewer's comment: In the line 176 should be: “breast milk is insufficient …”.
The authors' answer: We have corrected it.
The Reviewer's comment: The sentence: “patients often have low serum Se levels. content.” should be improved (line 342-343).
The authors' answer: We have corrected it.
Phenylketonuria diets are low in Se products, with no Se-rich products such as fish, meat, eggs and wholegrains, therefore phenylketonuria patients often have low serum Se levels.
The Reviewer's comment: The units in table 5 should be standardized: “µg/d” or “µg/day” or “per day …. µg” or “µg Se/day”; “mcg/L” or “µg/L” or “µg Se/L”.
The authors' answer: We have corrected it. We have marked the corrections in red in the manuscript.
The Reviewer's comment: “µg/l” should be corrected.
The authors' answer: We have corrected it. We have marked the corrections in red in the manuscript.
The Reviewer's comment: For a better comparison, it should be considering converting micromoles and also µg/dl.
The authors' answer: We have corrected it. We have marked the corrections in red in the manuscript.
The Reviewer's comment: In the same table please check in Caswell et al. should be “mg”??
The authors' answer: : It was an editorial mistake. We have corrected it. There should be “µg”.
The Reviewer's comment: Similarly, in Daniles et al. 3 g/kg/day is correct??
The authors' answer: It was an editorial mistake. We have corrected it. There should be “µg”.

Reviewer 2 Report
Comments and Suggestions for Authors
Even though the idea sounds interesting, the manuscript requires some improvements.
Although in the summary the authors say that this review evaluated studies... in the text, the authors do not adequately develop (lines 46-55) either what methods they used or what results they found. It appears to be more of a literature review than a critical review. The authors make a clear description of the knowledge (I assume current) about Selenium in humans (function, deficiency and excess, requirements in infants and children, main sources, its bioavailability, restrictive diets). Only in the last sections (6 - 8) do the authors refer to the limited availability of studies on this topic, without mentioning how many studies they found, the type of studies, where they were carried out, the meaning of their results, were the serum levels normal, low, high? etc. In this context, the authors should improve this manuscript.
The title should be improved and reflect the objective of this manuscript, for example, “Selenium in infants and preschool children nutrition: a literature review”.
Since what year was the search carried out? Is this correct what the authors write “AIe”? “Is”? …in other countries, for example? Is Table 6? Right?

Comments on the Quality of English LanguageMinor.
Author Response
Response to the comments made by the reviewers
Manuscript ID: nutrients-2666192
Title: Selenium in infants and preschool children nutrition, threat of excess or deficiency? A critical review
We would like to thank the Reviewers for their careful review of our manuscript and for providing us with some suggestions to improve its quality. We have carried out a major revision of the manuscript, and we believe the paper has improved significantly.
According to the Reviewers' suggestion, the manuscript has been carefully checked and corrected. The changes in the manuscript have been highlighted in red.
Below we sequentially address all of the points raised by the Reviewers.
Reviewer 2:
Firstly, we would like to express our profound thanks to the Reviewer for devoting time to reviewing our manuscript, the corrections and suggestions. We have carried out a major revision of the manuscript, and we believe the paper has improved significantly.
Major comments:
The Reviewer's comment: Although in the summary the authors say that this review evaluated studies... in the text, the authors do not adequately develop (lines 46-55) either what methods they used or what results they found. It appears to be more of a literature review than a critical review. The authors make a clear description of the knowledge (I assume current) about Selenium in humans (function, deficiency and excess, requirements in infants and children, main sources, its bioavailability, restrictive diets). Only in the last sections (6 - 8) do the authors refer to the limited availability of studies on this topic, without mentioning how many studies they found, the type of studies, where they were carried out, the meaning of their results, were the serum levels normal, low, high? etc. In this context, the authors should improve this manuscript.
The title should be improved and reflect the objective of this manuscript, for example, “Selenium in infants and preschool children nutrition: a literature review”.
Since what year was the search carried out? Is this correct what the authors write “AIe”? “Is”? …in other countries, for example? Is Table 6? Right?
The authors' answer: According to the Reviewer's suggestion, we have modified article.
- We have changed title of manuscript on “Selenium in infants and preschool children nutrition: a literature review”.
- We have clarification study design (added search scheme).
Figure 1. Search strategy.
- Search (period) - all available studies were included in the analysis, without time limits. Theoretically, the oldest work in the Scopus database came from 1973, but after narrowing the search with exclusion terms, 18 manuscripts were finally included, 5 of which came from before 2000 (one work from 1982, 1995, 1996, 1998 and 1999, respectively).
- We have added missing information about serum Se concentration, and added reference from a healthy pediatric population for serum Se concentrations. Table 6 (in new version Table 8) has been checked and modified.
- Serum Se concentration in infant and preschool children - overview of available studies
Table 8 presents the available studies examining the serum Se concentrations in infant and preschool children. The Se concentration in serum varied according to area, with Se deficiency in children from Malawi and a Se excess in children from Brazil [18,165,168]. This implies that the food consumed on a daily basis in a given region has a significant impact on the children’s Se status. Serum Se levels could be increased by the biofortification of a widely consumed food product [169].
For comparison, age-specific reference intervals based on the 2.5 and 97.5 percentiles of data derived from a healthy pediatric population for Se concentrations in serum are presented in Table 7 [170]
There have also been several results describing Se concentrations in newborns from birth through the first period of life, showing that the Se concentration is low and varies depending on the months life [18,165,170–172]. This can be easily modified by administering supplements enterally. However, it is important to select the appropriate dose and form, which affects its bioavailability. Nonetheless, the most effective way to maintain the correct level of Se in children is breastfeeding [173,174].
Table 6. Reference intervals (2.5 and 97.5 percentiles) from a healthy pediatric population for serum Se concentrations [170].
Age |
Serum Se concentrations (µg/L) |
< 1 month |
15-107 |
1-2 months |
15-100 |
2-4 months |
10-93 |
4-12 months |
13-116 |
1-5 years |
34-129 |
The Reviewer's comment: Is this correct what the authors write “AIe”? “Is”? …in other countries, for example? Is Table 6? Right?
The authors' answer: We have corrected editorial mistakes.
The estimated Se adequate intake of infants from birth is 12 µg/day and increasing with age to 20 µg/day for children aged 4−6.
However, in some cases, the Se content in breast milk is insufficient [80,81], therefore, future research should verify whether infants meet the recommended Se intake and assess the influence of the concurrent diet of lactating mothers on the Se content of their milk, especially in mothers on a diet poor in Se-rich products.
Depending on the region, the problem of Se intake is varied. In Brazil, children consumed so much Se that the amounts were considered potentially toxic, whereas the intake was insufficient in Philippines and Malavi [18,164,165].
- We have marked other corrections in red in the manuscript

Reviewer 3 Report
Comments and Suggestions for Authors
This is a fine and comprehensively written review on the problems of inadequate selenium intake, with particular focus on infants and pre-school children. Nevertheless, there are a few issues to be addressed for revision.
Major:
1. Provide data on normal serum concentrations, and critically low or high values within the text.
2. Replace the term 'content' for serum or plasma by the term 'concentration, as content depends on concentration and serum volume.
3. Frequently, the high variability of Se content in different foods, depending on origin or commercial product categories (like formula) is mentioned. However, this is of no practical help to readers, if for instance the content in Se of wheat, other crops eggs, fish etc is not specified with respect to origin (e.g. imported wheat from Northamerica vs. grown in different European countries).
4. Transfer Se concentrations in foods into wet weight concentrations throughout the manuscript, including tables.
5. L.312: Provide values of bioavailability of organic vs. inorganic compounds. This is important for the evaluation of Se supplements with anorganic Se.
6. Conclusions, l. 426. What about developmental disorders, hypothyreosis etc? The authors might extend the issues of particular importance of Se deficiency in children.
Minor:
1. L. 195: Replace determination by 'quantifcation'.
2. L. 225: What is a safe source? Please provide location of origin and the reason for heat treatment under this circumstance.
3. L. 248 µg rather than g!
4. L. 254: Dairy products are not an essential calcium source. What abour vegane milk replacements?
5. L. 255: Specify product concentrationsand variability.
6. L259: 'consumption' of what, milk or of selenium?
7. L269f: Yes, but what does this mean for application? Specify the adequate regions and local conditions to be translated into life. Moreover, thransfer data to wet weight uniformly in the paper.
8. L274: What does this mean? 100g onion in a total meal of a 6y old child is about 50% of AI then. A translation of recommendation into practical life would increase perception of data!
9. L289f: Provide ranges in Brasil nuts, and the values in different regions, if different and possible.
10. L. 292: Provide an explanation, at least for barium toxicity.
11. L 302: Is this study by Adams based on local analyses? Does it apply to European compared to North American Se concentrations of wheat similarly?
12. L. 308f: Provide data, how much yeast is added to important regular food items.
13. L 310ff: Sum this up in form of a table. Explain the respective different forms of carbohydrates/polysaccharides affecting bioavailability.
14. L. 330: Are there other vitamins than A,D&E increasing bioavailability. Otherwise write: ...in the presence of viatamins A, D and E...
15: L. 334f: What does this sentence mean? Please explain or rephrase.
16. L 342f: Provide values of 'low serum levels' compared to controls.
17. L. 353: Euthyroid status is not defined by thyrotropin level alone, but by (f)T4/T3 and other clinical parameters.
18: L. 357: Please specify parameters being addressed in these allergic children.
19: L. 375: Specify countries namely, please.
20: L. 379f: Provide names, please, of the good and bad, with year of analysis.
21: L. 390: The medical importance of such postnatal decrease is uncertain, as it may be physiologic to term infants. Question is only, whether it reincreases accoring to adequate provision by breast milk (or formula), and what it implies for preterm infants, having such decrease at an inadequate time point.
22. Table 5: Please specify for each study, e.g. in a separate column, whether studies were retro-/prospective, observational/interventional, blinded, randomized, controlled!
Specify, whether data are means and standard deviations, medians and IQR or range.
23. Table 5, Philippines: 'inadequately low' rather than inadequate?
24: Table 6, heading: replace content by concentration.
25. Table 6, STrauss et al, 2010, >Supplementation of 4-9 μg/kg/d raised plasma Se levels, but they remained lower than in the healthy poplulation<: Please provide concentration values.
26. Table 7, line 3: Replace >content< by concentration. Line 6: Provide product names for those at danger of intake above UL. Line 7: Define the Se-rich products.
27: L. 416f: Provide ranges and possible origins of high and low Se Brasil nuts, as far as possible from the reference.
Author Response
Response to the comments made by the reviewers
Manuscript ID: nutrients-2666192
Title: Selenium in infants and preschool children nutrition, threat of excess or deficiency? A critical review
We would like to thank the Reviewers for their careful review of our manuscript and for providing us with some suggestions to improve its quality. We have carried out a major revision of the manuscript, and we believe the paper has improved significantly.
According to the Reviewers' suggestion, the manuscript has been carefully checked and corrected. The changes in the manuscript have been highlighted in red.
Below we sequentially address all of the points raised by the Reviewers.
Reviewer 3:
Firstly, we would like to express our profound thanks to the Reviewer for devoting time to reviewing our manuscript, the corrections and suggestions. We have carried out a major revision of the manuscript, and we believe the paper has improved significantly.
Major comments:
The Reviewer's comment: Provide data on normal serum concentrations, and critically low or high values within the text.
The authors' answer: According to the Reviewer's suggestion, we have added information about serum Se concentration from healthy pediatric population.
For comparison, age-specific reference intervals based on the 2.5 and 97.5 percentiles of data derived from a healthy pediatric population for Se concentrations in serum are presented in Table 7 [170].
Table 7. Reference intervals (2.5 and 97.5 percentiles) from a healthy pediatric population for serum Se concentrations [170].
Age |
Serum Se concentrations (µg/L) |
< 1 month |
15-107 |
1-2 months |
15-100 |
2-4 months |
10-93 |
4-12 months |
13-116 |
1-5 years |
34-129 |
The Reviewer's comment: Replace the term 'content' for serum or plasma by the term 'concentration, as content depends on concentration and serum volume.
The authors' answer: We have corrected it. We have marked the corrections in red in the manuscript.
The Reviewer's comment: Frequently, the high variability of Se content in different foods, depending on origin or commercial product categories (like formula) is mentioned. However, this is of no practical help to readers, if for instance the content in Se of wheat, other crops eggs, fish etc is not specified with respect to origin (e.g. imported wheat from Northamerica vs. grown in different European countries).
The authors' answer: According to the Reviewer's suggestion, we have added more information about Se concentration depending on origin (Brazil nuts, fish, milk and milk pruducts).
…It is worth noting, the Se concentration in food products varies and depends on the origin and culinary processing [77]….
For Brazil nuts: The Se concentration of Brazil Nuts from Brazilian Amazon basin are varied from in Mato Grosso state (2.4 µg/g) and in Acre state (3.0 µg/g) to in Amazonas state (66.1 µg/g) and in Amapa state (51.2 µg/g) [128].
For fish: Additionally, Se concentration in fish is varies depends on the origin e.g. Se concentrations in tuna from Spain and Portugal are 0.92±0.01 µg/g [98], from New Jersey 0.43±0.04 µg/g [99], and form Japan 0.75 µg/g [100].
For milk and milk products: According to Yanardag et al., good Se sources are various cheeses (0.02-0.29 µg/g), butter (0.03-0.22 µg/g), coffee cream (0.03-0.25 µg/g), and milk powder (0.06-0.10 µg/g).
We agree with the Reviewer that providing information on the Se content in products depending on their origin would be very valuable for a potential reader (especially since the Se content in wheat products may range from 15–2372 μg kg–1 in depending on origin). However, consumers are often not aware of where the wheat from which the breakfast cereals, bread and pasta they eat comes from. All the more so because data published by the European Commission shows that the largest number is durum wheat, as many as 2.3 million. tons, or 71% in the 2020/2021 season so far came from Canada. The United States is second with a share of 16%. The proportions are different in the case of soft wheat; most of this comes to the EU from Ukraine - 1.73 million tons - which gives a share of 35%. Next are Canada, Russia, the USA and Serbia with shares of 26, 15, 11 and 8% respectively.
Therefore, we have not provided detailed information on the data provided in the article.
Wang M, Li B, Li S, Song Z, Kong F, Zhang X. Selenium in Wheat from Farming to Food. J Agric Food Chem. 2021 Dec 29;69(51):15458-15467. doi: 10.1021/acs.jafc.1c04992.
The Reviewer's comment: Transfer Se concentrations in foods into wet weight concentrations throughout the manuscript, including tables.
The authors' answer: According to the Reviewer's suggestion, we have corrected Se concentrations. However, in Se-rich garlic and Se-rich onion we cant find this information. We have marked the corrections in red in the manuscript.
The Reviewer's comment: L.312: Provide values of bioavailability of organic vs. inorganic compounds. This is important for the evaluation of Se supplements with anorganic Se.
The authors' answer: According to the Reviewer's suggestion, we have added information about bioavailability values.
As previously mentioned, organic compounds are more bioavailable than inorganic (selenomethionine and selenosysteine >90%, selenate and selenite >50% absorbed).
The Reviewer's comment:. Conclusions, l. 426. What about developmental disorders, hypothyreosis etc? The authors might extend the issues of particular importance of Se deficiency in children.
The authors' answer: We have added missing information.
Selenium deficiency is very common in areas with low selenium soil, including highly developed countries in most of Europe. Its may contribute to developmental disorder, hypothyroidism and lowering immune response. Therefore, screening of the Se content in infants and children in the context of possible deficiencies may be neces-sary. This will be of particular importance in children with an increased risk of defi-ciency, i.e. patients with phenylketonuria or food allergies. However, due to the awareness of possible selenium deficiency, it is worth monitoring the levels in patients with symptoms of deficiency, e.g. thyroid problems.
Minor:
The Reviewer's comment:. L. 195: Replace determination by 'quantifcation'.
The authors' answer: We have corrected it.
The Reviewer's comment: L. 225: What is a safe source? Please provide location of origin and the reason for heat treatment under this circumstance.
The authors' answer: According to the Reviewer's suggestion, we have added missing information.
Therefore, fish from a verified source and after prior heat treatment (reducing the pathogenic microorganism) should only be served [101].
The Reviewer's comment: L. 248 µg rather than g!
The authors' answer: We have corrected it.
The Reviewer's comment: L. 254: Dairy products are not an essential calcium source. What abour vegane milk replacements?
The authors' answer: We have corrected it.
The Se content in milk and dairy products is much lower (Table 4) but since they are a natural source of calcium, their consumption in humans should be increased.
The Reviewer's comment: L. 255: Specify product concentrations and variability.
The authors' answer: We have added missing information.
According to Yanardag et al., good Se sources are various cheeses (0.02-0.29 µg/g), butter (0.03-0.22 µg/g), coffee cream (0.03-0.25 µg/g), and milk powder (0.06-0.10 µg/g).
The Reviewer's comment: L259: 'consumption' of what, milk or of selenium?
The authors' answer: Consumption of selenium. We have corrected it.
The Reviewer's comment: L269f: Yes, but what does this mean for application? Specify the adequate regions and local conditions to be translated into life. Moreover, thransfer data to wet weight uniformly in the paper.
The authors' answer: We have corrected this information.
It has been shown that Brassicaceae family can accumulate up to 150 fold more Se in their tissues when grown in Se-rich soil (e.g. Brassica grains collected in normal soli 1,2±0.12 µg/g, Brassica grains collected from seleniferous soil 183±25.3 µg/g). The Se concentrations in broccoli, brussels sprouts and cabbage grown in not Se-enriched soil are 0.01-0.03 µg/g, 0.004-0.06 µg/g 0.001-0.02 µg/g respectively [113].
The Reviewer's comment: L274: What does this mean? 100g onion in a total meal of a 6y old child is about 50% of AI then. A translation of recommendation into practical life would increase perception of data!
The authors' answer: We have added missing information about practical use, but unfortunately we can not find any information about se-rich garlic and Se-rich onion in wet mass.
Allium vegetables are a good source of selenium, although due to limited consumption by young childrenb it may not significantly affect the selenium content in their diet.
The Reviewer's comment: L289f: Provide ranges in Brasil nuts, and the values in different regions, if different and possible.
The authors' answer: We have added this information.
The Se concentration of Brazil Nuts from Brazilian Amazon basin are varied from in Mato Grosso state (2.4 µg/g) and in Acre state (3.0 µg/g) to in Amazonas state (66.1 µg/g) and in Amapa state (51.2 µg/g) [127].
The Reviewer's comment: L. 292: Provide an explanation, at least for barium toxicity.
The authors' answer: According to the Reviewer's suggestion, we have added missing information.
Of note, the toxic values of barium (sub-acute exposure to Ba can cause muscle weak-ness) [130,131] and radium taken with food are not clearly defined but barium toxicity has been reported with ingestions as small as 200 mg Ba/kg/day [132,133].
The Reviewer's comment: L 302: Is this study by Adams based on local analyses? Does it apply to European compared to North American Se concentrations of wheat similarly?
The authors' answer: We have added missing information.
Interestingly, Adams et al. found no long-term changes in the distribution of Se concentrations in wheat grains over 17 years regions around the United Kingdom [134].
The Reviewer's comment: L. 308f: Provide data, how much yeast is added to important regular food items.
The authors' answer: According to the Central Statistical Office data, yeast consumption is not high and amounts to approximately 5g/day. Therefore, a statement was added in the manuscript - yeast consumption is low.
(https://stat.gov.pl/files/gfx/portalinformacyjny/pl/defaultaktualnosci/5515/5/15/1/rocznik_statystyczny_przemyslu_2021.pdf)
The Se content in bakery products can also be increased by adding yeast, as the commercial dried product contains 2.5 mg Se/g (range 1−2.4 mg Se/g), however, yeast consumption is low [140].
The Reviewer's comment: L 310ff: Sum this up in form of a table. Explain the respective different forms of carbohydrates/polysaccharides affecting bioavailability.
The authors' answer: According to the Reviewer's suggestion, we have added a table. According to other Reviewer we have removed information about polysaccharides in mushrooms - this does not apply to feeding small children
Table 5. Bioavailability of Se depending on food ingredients.
Food ingredients |
Effect on bioavailability |
Proteins |
↓ Se bioavailability with increasing protein content in fish and seafood |
Fats |
↑ Se bioavailability with increase polyunsaturated fatty acid in diet – animal model study ↑ Se bioavailability with reducing the fat content in milk |
Carbohydrates |
↑ Se bioavailability – analyzed in fish and seafood |
Dietary fibre |
↓ Se bioavailability |
Vitamins A, D and E |
↑ Se bioavailability |
Sulphur |
↓ Se bioavailability sulfur in diet may compete with Se for absorption |
The Reviewer's comment: L. 330: Are there other vitamins than A,D&E increasing bioavailability. Otherwise write: ...in the presence of viatamins A, D and E...
The authors' answer: . We have corrected it.
Se is most easily absorbed in the presence of vitamins A, D and E can increase Se bioavailability, while heavy metals (especially mercury) and fibre decrease it [151,152].
The Reviewer's comment: L. 334f: What does this sentence mean? Please explain or rephrase.
The authors' answer: We have corrected it.
Additionally, parameters related to the human body, i.e., Se status, age, sex, or life-style factors may influence on bioavailability of Se [139].
The Reviewer's comment: L 342f: Provide values of 'low serum levels' compared to controls.
The authors' answer: According to the Reviewer's suggestion, we have added missing information.
In Okano et al. study, the serum Se concentration in all analyzed patients with PKU (n=11) in age 4-38 years was lower than the reference value (106–174 µg/dL) and amounted to 56.6 ± 21.2 µg/L [153].
The Reviewer's comment: L. 353: Euthyroid status is not defined by thyrotropin level alone, but by (f)T4/T3 and other clinical parameters.
The authors' answer: We have corrected this sentence.
Moreover, according to Chanoine study, in patients with isolated Se deficiency (e.g., in patients with phenylketonuria on a low-protein diet), the metabolism of peripheral thyroid hormones was disturbed, but no changes in thyrotropin concentration or clin-ical symptoms of hypothyroidism were observed, which suggests that these patients are euthyroid. Although this statement may be questionable because euthyroidism should be defined on the basis of (f)T4/T3 and other clinical parameters, it is suggested that Se supplementation is recommended for patients following a diet low in natural protein.
The Reviewer's comment: L. 357: Please specify parameters being addressed in these allergic children.
The authors' answer: We have added missing information.
Recently, attention has been paid to the occurrence of Se deficiency in food allergic children (IgE-mediated).
The Reviewer's comment: L. 375: Specify countries namely, please.
The authors' answer: We have added missing information.
In Brazil, children consumed so much Se that the amounts were considered potentially toxic, whereas the intake was insufficient in Philippines and Malavi [18,161,162].
The Reviewer's comment: L. 379f: Provide names, please, of the good and bad, with year of analysis.
The authors' answer: We have added missing information.
Research also shows that some infant milk formulas do not contain the right amounts of Se to ensure adequate daily intake, however these studies are from 1982 [164]. Currently, the Se content in infant milk formula is strictly defined by the Delegated Regulation (EU) [83,84].
The Reviewer's comment: L. 390: The medical importance of such postnatal decrease is uncertain, as it may be physiologic to term infants. Question is only, whether it reincreases accoring to adequate provision by breast milk (or formula), and what it implies for preterm infants, having such decrease at an inadequate time point.
The authors' answer: We have modified this sentence.
There have also been several results describing Se concentrations in newborns from birth through the first period of life, showing that the Se concentration is low and varies depending on the months life [18,162,167–169].
The Reviewer's comment: Table 5: Please specify for each study, e.g. in a separate column, whether studies were retro-/prospective, observational/interventional, blinded, randomized, controlled!
Specify, whether data are means and standard deviations, medians and IQR or range.
The authors' answer: According to the Reviewer's suggestion, we have added information about type of study in separate column and information about data.
The Reviewer's comment: Table 5, Philippines: 'inadequately low' rather than inadequate?
The authors' answer: We have corrected it.
The Reviewer's comment: Table 6, heading: replace content by concentration.
The authors' answer: We have corrected it.
The Reviewer's comment: Table 6, STrauss et al, 2010, >Supplementation of 4-9 μg/kg/d raised plasma Se levels, but they remained lower than in the healthy poplulation<: Please provide concentration values.
The authors' answer: We have added missing information.
The Reviewer's comment: Table 7, line 3: Replace >content< by concentration. Line 6: Provide product names for those at danger of intake above UL. Line 7: Define the Se-rich products.
The authors' answer: We have corrected it.
Table 9. Dietary Se recommendations for infants and young children
Infants |
· Breastfeeding is recommended until the age of 6 months, however, the diet of breastfeeding women should include products rich in Se. Nonetheless, no studies confirming the correlation between the Se concentration in the mother's diet and the Se content in the milk have been published [27,78–80,176]. |
· In the case of infants who cannot be breastfed, it is recommended to use a milk formula (Se content regulated by the EU and FDA). However, in infants with increased energy demand and the use of milk with the highest permitted Se concentration (8.6 µg/100 kcal) and feeding exclusively with this milk (in infants over 4 months), the UL value may be exceeded [86]. |
Expanding diet |
· Introduce Se-rich products into the child's diet: meat, fish, eggs, cereals, Brassica vegetables · For children with allergies, products with a high Se content, i.e. fish and meat should be replaced with other Se-rich products or consider supplementation · It is recommended to monitor serum Se and introduce Se supplementation in children with phenylketonuria [54,154,159,159]. |
Dietary recommendation from 1 to3 years old |
· The diet should be varied and rich in products containing Se · Children should not consume Brazil nuts in excess (1 Brazil nut has around 50 µg )* · Se-containing supplements in toddlers and children should be used with caution, based on individual needs [18,69,86]. |
The Reviewer's comment: L. 416f: Provide ranges and possible origins of high and low Se Brasil nuts, as far as possible from the reference.
The authors' answer: We have added this information.
The Se concentration of Brazil Nuts from Brazilian Amazon basin are varied from in Mato Grosso state (2.4 µg/g) and in Acre state (3.0 µg/g) to in Amazonas state (66.1 µg/g) and in Amapa state (51.2 µg/g) [127].

Round 2
Reviewer 2 Report
Comments and Suggestions for Authors
Even though the idea of this review sounds interesting. There are some important points that need to be improved.
Abstract: Reformulate the main objective of this literature review. Authors should not use the words that appear in the title as keywords.
Introduction: It would be better if the authors tried to be more specific: the introduction section should guide readers to understand why they undertook this review. It would be a good idea for the authors to formulate a hypothesis and then describe the main goal of this literature review, for example, “We believe that there is a risk of Se deficiency in infants and schoolchildren related to their daily consumption. Therefore, the main objective of this literature review was to summarize what is known to date about Se levels and the risk of deficiency related to regular consumption in infants and schoolchildren.”
It would be better if the authors added a subsection describing the methodology they used to collect the articles related to their objective. It would be better to add “All available studies were included in the analysis, without time limits. Theoretically since 1973, but after narrowing the search with exclusion terms, 18 manuscripts were finally included, 5 of which were before 2000 (one work from 1982, 1995, 1996, 1998 and 1999, respectively).” Delete what is crossed out. What about Se levels? What data was collected? All variables collected must be described in detail: age, age group, sex, demographic characteristics, dietary intake, dietary Se sources, maternal diet, breast milk, infant formulas, recommendation guidelines, Se bioavailability, Se deficiency, serum levels, reference, Se. levels. , normal cut-off points, etc. It would be better to write Se's full name in all titles and subtitles. Tables 3 and 6 are not described in the text. Is this the normal range of Se levels for all ages? Specify. What are the symptoms of Se deficiency? Is this "(f)" correct? Up to now, there is .... (describe them). What happens in other countries/regions? How many studies? Specify. Table 6? Are these low levels in poor ranges? Specify. Based on what data? Please specify more. References. For example… What the authors have written in the conclusion seems to be an argument for discussion. It would be interesting if the authors, after all the information provided, wrote a short discussion to highlight the most important points and end with a conclusion. The authors should focus their discussion on the central theme of the study and avoid repeating the information given. The authors should indicate why this review is important and to what extent it contributes to current knowledge of this topic. A paragraph of limitations and suggestions for this review should be written before the conclusion.

Author Response
Response to the comments made by the Reviewer
Manuscript ID: nutrients-2666192
Title: Selenium in infants and preschool children nutrition, threat of excess or deficiency? A critical review
New title: Selenium in infants and preschool children nutrition: a literature review
Dear Reviewer, we appreciate all your insightful comments. Thank you for your suggestions.
The Reviewer's comment: Abstract: Reformulate the main objective of this literature review. Authors should not use the words that appear in the title as keywords.
The authors' answer: According the Reviewer’s suggestion, we have corrected abstract and keywords.
Abstract: Selenium (Se), an essential trace element, is fundamental to human health, playing an important role in the formation of thyroid hormones, DNA synthesis, the immune response and fertility. There is a lack of comprehensive epidemiological research, particularly the serum Se con-tent in healthy infants and preschool children compared to the estimated dietary Se intake. How-ever, Se deficiencies and exceeding the UL have been observed in infants and preschool children. Despite the observed irregularities in Se intake, there is a lack of nutritional recommendations for infants and preschool children. Therefore, the main objective of this literature review was to summarize what is known to date about Se levels and the risk of deficiency related to regular consumption in infants and preschool children.
The Reviewer's comment: Introduction: It would be better if the authors tried to be more specific: the introduction section should guide readers to understand why they undertook this review. It would be a good idea for the authors to formulate a hypothesis and then describe the main goal of this literature review, for example, “We believe that there is a risk of Se deficiency in infants and schoolchildren related to their daily consumption. Therefore, the main objective of this literature review was to summarize what is known to date about Se levels and the risk of deficiency related to regular consumption in infants and schoolchildren.”
The authors' answer: We have corrected it.
We believe that there is a risk of Se deficiency in infants and preschool children related to their daily consumption. Therefore, the main objective of this literature re-view was to summarize what is known to date about Se levels and the risk of deficiency related to regular consumption in infants and preschool children.
The Reviewer's comment: It would be better if the authors added a subsection describing the methodology they used to collect the articles related to their objective. It would be better to add “All available studies were included in the analysis, without time limits. Theoretically since 1973, but after narrowing the search with exclusion terms, 18 manuscripts were finally included, 5 of which were before 2000 (one work from 1982, 1995, 1996, 1998 and 1999, respectively).”
Delete what is crossed out. What about Se levels? What data was collected? All variables collected must be described in detail: age, age group, sex, demographic characteristics, dietary intake, dietary Se sources, maternal diet, breast milk, infant formulas, recommendation guidelines, Se bioavailability, Se deficiency, serum levels, reference, Se. levels. , normal cut-off points, etc.
The authors' answer: We have corrected it. We have added subsection and missing information.
1.1. Methodology
Relevant articles regarding Se in the diet of infants and children were retrieved from PubMed, Scopus, Web of Science, Embase and Cochrane Library using the fol-lowing keywords: “selenium” and “food products” and “healthy children” or “sele-nium” and “food products” and “healthy infants". An additional search was per-formed for selenium content” and “healthy children” or “selenium content” and “healthy infants”. Duplicate articles and studies in languages other than English or adults were excluded.
Only studies on infant and preschool children (age: birth-5 years) were included in the analysis. The analysis took into account the type of study, country, age group, gender, demographic characteristics, Se intake, sources of Se in the diet, and serum Se levels.
All available studies were included in the analysis, without time limits. Theoretically since 1973, but after narrowing the search with exclusion terms, 18 manuscripts were finally included, 5 of which were before 2000 (one work from 1982, 1995, 1996, 1998 and 1999, respectively). The reference lists of retrieved articles were screened manually to find potential relevant literature (Figure 1).
The Reviewer's comment: It would be better to write Se's full name in all titles and subtitles. Tables 3 and 6 are not described in the text. Is this the normal range of Se levels for all ages? Specify. What are the symptoms of Se deficiency? Is this "(f)" correct? Up to now, there is .... (describe them). What happens in other countries/regions? How many studies? Specify. Table 6? Are these low levels in poor ranges? Specify. Based on what data? Please specify more. References. For example…
The authors' answer: We have corrected it. The changes in the manuscript have been highlighted in red.
- Is this the normal range of Se levels for all ages? – In Okano et al. study was not detailed this information, therefore the reference values were removed from the manuscript.
In Okano et al. study, the serum Se concentration in all analyzed patients with PKU (n=11) in age 4-38 years was lower than the reference value (106–174 µg/L) and amounted to 56.6 ± 21.2 µg/L [156].
- What are the symptoms of Se deficiency? - The author of this study does not indicate specific symptoms (test criteria) of Se deficiency.
- Up to now, there is .... Up to now, there are three studies assessing Se intake in preschool children [18,166,167] and three studies in infants [166,168,169] (Table 7).
- Are these low levels in poor ranges? Specify. - In most of analyzed studies, mean Se concentrations in serum were low, but well within the reference intervals shown in Table 6. Only Zyambo et al. study had lower values, but it was within group of children with severe acute malnutrition [171].
The Reviewer's comment: What the authors have written in the conclusion seems to be an argument for discussion. It would be interesting if the authors, after all the information provided, wrote a short discussion to highlight the most important points and end with a conclusion. The authors should focus their discussion on the central theme of the study and avoid repeating the information given. The authors should indicate why this review is important and to what extent it contributes to current knowledge of this topic. A paragraph of limitations and suggestions for this review should be written before the conclusion.
The authors' answer: According to the Reviewer's suggestion, we have added a new discussion section to which we have added the strengths and weaknesses of the study. However, due to its shortened form, we called this paragraph “Summary”.
- Summary
The main focus of this article is to provide an all-encompassing account of the significance of selenium in the diet of young children (infants and preschoolers). It is important to note that existing literature on selenium can be categorized into two distinct groups: (1) descriptions of dietary sources, bioavailability, physiological functions, and deficiency symptoms, and (2) cross-sectional examinations of selenium levels in particular populations. Our research takes a comprehensive approach to this issue. The population under study, namely children aged up to 5-6 years, is of particular significance due to two seemingly opposing reasons: these children are at higher risk of selenium deficiency owing to food allergies or selective feeding, while, conversely, they may also suffer from overdosage due to smaller requirements and food fortification. To the best of our knowledge, this is the first piece of research to concentrate specifically on the difficulties encountered by this group. However, this study's scope is constrained by the absence of a meta-analysis, which precludes statistical interpretation of the aggregated results.
Currently, there is a lack of epidemiological studies regarding serum selenium content, especially compared with estimated dietary intake of selenium in infants and preschool children. Therefore, there is a lack of nutritional strategies for infants and young children regarding selenium intake.
Selenium deficiency is very common in areas with low selenium soil, including highly developed countries in most of Europe. Its may contribute to developmental disorder, hypothyroidism and lowering immune response. Therefore, screening of the Se content in infants and children in the context of possible deficiencies may be necessary. This will be of particular importance in children with an increased risk of deficiency, i.e. patients with phenylketonuria or food allergies. However, due to the awareness of possible selenium deficiency, it is worth monitoring the levels in patients with symptoms of deficiency, e.g. thyroid problems.
- Conclusion
Due to possible Se deficiencies in infants and preschool children, epidemiological studies are needed to determine Se intake and serum Se concentrations in these population groups.

Reviewer 3 Report
Comments and Suggestions for Authors
The authors have adequately revised theire manuscript, and it's fine paper now, providing much more detailed information and reflection of clinical conetxt. There remain a few minor comments:
1. Table 3: The units for breast milk and formula are not comparable. Adapt breast milk values to the same unit as done for formula (e.g. by using mean energy content of breast milk).
2. Table 4: Provide data in terms of regular intake/meal size. This table afford much calculation. Who konows, what an egg york weights? Moreover, values don't provide ranges!
3. L. 219f: Grammar: ...varies, and depends...
4. L. 251: >The Se concentration in meat varies...<: So, why do you not show that in the table?
5. L. 264: ...good Se sources: ? The data say: may be good sources, if from the right origin!
6. L. 280: Who eats Brassic grains on a regular basis?
7. L303: >are varied<: grammar: vary
8. L. 307: >the high Se content<: This is contradictory to lines 303ff, showing that the area of origin is important, and content is highly variable. Rephrase, please.
9. L. 337: Grammar: affecting, rather than affect.
10: L 351f: Grammar of sentence! Rephrase, please.
11: L366: Grammar. Start new sentence with >Therefore<.
Comments on the Quality of English LanguageOnly minor corrections required. Otherwise, English is fine.
Author Response
Response to the comments made by the Reviewer
Manuscript ID: nutrients-2666192
Title: Selenium in infants and preschool children nutrition, threat of excess or deficiency? A critical review
New title: Selenium in infants and preschool children nutrition: a literature review
Dear Reviewer, we appreciate all your insightful comments. Thank you for your suggestions.
The Reviewer's comment: 1. Table 3: The units for breast milk and formula are not comparable. Adapt breast milk values to the same unit as done for formula (e.g. by using mean energy content of breast milk).
The authors' answer: According to Reviewer’s suggestion, we have modified data in Table 3.
Table 3. Se content in best milk and infant formulae
Food source |
Average content |
Comments |
Breast milk |
2.2–3.0 μg/100 kcal* |
Se content depends on the maternal diet |
Infant formulae |
3.0−8.6 μg /100 kcal |
According to the Delegated Regulation (EU) |
|
2.0−7.0 μg /100 kcal |
According to the FDA |
* taking into account that 100 ml of breast milk has 67 kcal [90]
The Reviewer's comment: 2. Table 4: Provide data in terms of regular intake/meal size. This table afford much calculation. Who konows, what an egg york weights? Moreover, values don't provide ranges!
The authors' answer: We have corrected this table.
Table 4. Main Se sources in children aged 1 year and older
Food source |
Average content µg/g |
Average Se content µg/per serving [serving size] |
Fish [75] |
0.4−4.3 |
20–215 µg [50 g] |
Meats (mussels) [75] |
0.03–0.45 |
1.5–22.5 µg [50 g] |
Yolk from egg [92] |
0.12−0.42 |
1.2–4.2 µg [10 g – ½ piece} |
Cereals [93] |
0.01-0.55 |
0.75– 41.25 µg [75 g] |
Broccoli [94] |
0.02 |
1 µg [50 g] |
Cow’s milk [95] |
0.01−0.02 |
0.5– 1 µg [50 ml] |
Gouda cheese [95] |
0.08 |
1.2 µg [15 g – 1 one slice] |
Yoghurt [95] |
0.02 |
1 µg [50 g] |
The Reviewer's comment: 3. L. 219f: Grammar: ...varies, and depends...
The authors' answer: We have corrected it.
Additionally, Se concentration in fish depends on the origin e.g. Se concentrations in tuna from Spain and Portugal are 0.92±0.01 µg/g [99], from New Jersey 0.43±0.04 µg/g [100], and form Japan 0.75 µg/g [101].
The Reviewer's comment: 4. L. 251: >The Se concentration in meat varies...<: So, why do you not show that in the table?
The authors' answer: We have corrected it in the table.
The Reviewer's comment: 5. L. 264: ...good Se sources: ? The data say: may be good sources, if from the right origin!
The authors' answer: We have corrected it.
According to Yanardag et al., Se content in dairy products varies and depending on origin: in butter (0.03-0.22 µg/g), coffee cream (0.03-0.25 µg/g), cheeses (0.02-0.29 µg/g), and milk powder (0.06-0.10 µg/g).
The Reviewer's comment: 6. L. 280: Who eats Brassic grains on a regular basis?
The authors' answer: We have corrected it.
It has been shown that Brassicaceae family can accumulate up to 300-fold more Se in their tissues when grown in Se-rich soil. In Gui et al. study was evidenced Se contents in Broccoli florets were significantly higher under Se yeast treatments (300-fold) and under selenite treatments (50-fold) than under non-selenized treatments [117].
The Reviewer's comment: 7. L303: >are varied<: grammar: vary
The authors' answer: We have corrected it.
The Reviewer's comment: 8. L. 307: >the high Se content<: This is contradictory to lines 303ff, showing that the area of origin is important, and content is highly variable. Rephrase, please.
The authors' answer: We have corrected it.
However, due to the possible high Se content, children should not consume Brazil nuts in excess (1 Brazil nut contains approximately 50 µg) [69].
The Reviewer's comment: 9. L. 337: Grammar: affecting, rather than affect.
The authors' answer: We have corrected it.
The major food ingredients affecting bioavailability are carbohydrates, proteins, fat and dietary fibre, and minor food components include vitamins, toxic metals and oligoelements [148].
The Reviewer's comment: 10: L 351f: Grammar of sentence! Rephrase, please.
The authors' answer: We have corrected it.
Se is most easily absorbed in the presence of vitamins A, D and E. They increase Se bioavailability, while heavy metals (especially mercury) and fibre decrease it [155,156].
The Reviewer's comment: 11: L366: Grammar. Start new sentence with >Therefore<.
The authors' answer: We have corrected it.
Phenylketonuria diets are low in protein products, with no Se-rich products such as fish, meat, eggs and wholegrains. Therefore phenylketonuria patients often have low serum Se levels. In Okano et al. study, the serum Se concentration in all analyzed patients with PKU (n=11) in age 4-38 years was lower than the reference value (106–174 µg/L) and amounted to 56.6 ± 21.2 µg/L [157].
